# Dopamine neurons projecting to medial shell of the nucleus accumbens drive heroin reinforcement

Julie Corre[1], Ruud van Zessen[1], Michaël Loureiro[1], Tommaso Patriarchi[2], Lin Tian[2], Vincent Pascoli[1], Christian Lüscher[1,3]*

[1]Department of Basic Neurosciences, Medical Faculty, University of Geneva, Geneva, Switzerland; [2]School of Medicine, Department of Biochemistry and Molecular Medicine, University of California Davis, California, United States; [3]Service of Neurology, University of Geneva Hospital, Geneva, Switzerland

**Abstract** The dopamine (DA) hypothesis posits the increase of mesolimbic dopamine levels as a defining commonality of addictive drugs, initially causing reinforcement, eventually leading to compulsive consumption. While much experimental evidence from psychostimulants supports this hypothesis, it has been challenged for opioid reinforcement. Here, we monitor genetically encoded DA and calcium indicators as well as cFos in mice to reveal that heroin activates DA neurons located in the medial part of the VTA, preferentially projecting to the medial shell of the nucleus accumbens (NAc). Chemogenetic and optogenetic manipulations of VTA DA or GABA neurons establish a causal link to heroin reinforcement. Inhibition of DA neurons blocked heroin self-administration, while heroin inhibited optogenetic self-stimulation of DA neurons. Likewise, heroin occluded the self-inhibition of VTA GABA neurons. Together, these experiments support a model of disinhibition of a subset of VTA DA neurons in opioid reinforcement.
DOI: https://doi.org/10.7554/eLife.39945.001

*For correspondence:
Christian.Luscher@unige.ch

## Introduction

The DA hypothesis of drug reinforcement is rooted in the observation that electrical activation of the medial forebrain bundle leads to repetitive action (*Olds and Milner, 1954*). Rats willingly self-stimulate brain regions populated by DA neurons or receiving inputs from DA neurons. Moreover, pharmacological blockade of DA receptors impairs the reinforcing properties of psychostimulants in both rats (*Maldonado et al., 1993*; *McGregor and Roberts, 1993*; *Roberts et al., 1977*) and primates (*Bergman et al., 1989*; *Johanson and Schuster, 1975*). Several microdialysis and voltammetry studies demonstrated the increase of DA in the NAc shell as a common feature of addictive drugs, including opioids (*Aragona et al., 2008*; *Di Chiara and Imperato, 1988*; *Pontieri et al., 1995*; *Stuber et al., 2005*). Furthermore, electrolytic lesions of the VTA to NAc pathway decreased reinforcement during intravenous self-administration of morphine and cocaine under a progressive ratio schedule (*Suto et al., 2011*).

The DA hypothesis has also received support from molecular investigations. Indeed, the reinforcing effects of opioids require μ-opioid receptors (*Charbogne et al., 2017*; *Contarino et al., 2002*; *Matthes et al., 1996*), which are enriched in VTA GABA neurons (*Cohen et al., 1992*; *Devine and Wise, 1994*; *Johnson and North, 1992*). Based on in vivo single unit and brain slice recordings, a disinhibition *scenario* of VTA DA neurons has been proposed (*Gysling and Wang, 1983*), whereby MOR activation inhibits GABA neurons (*Johnson and North, 1992*) through somatodendritic hyperpolarization and the reduction of the efferent release probability. The former effect would be

mediated by G protein–coupled inwardly rectifying K+ (GIRK) channels, while inhibition of calcium entry underlies the later (*Lüscher et al., 1997*).

Regardless, it has been repeatedly argued that the initial reinforcing effects of opioids, can escape DA involvement. These results were largely based on pharmacological experiments. For example, the non-selective DA antagonists alpha-flupenthixol and haloperidol decreased cocaine SA but only to a lesser extent heroin SA (*Ettenberg et al., 1982*; *Van Ree and Ramsey, 1987*). In addition, lesioning DA terminals in the NAc with 6-OHDA had no effect on the initiation of heroin self-administration (*Gerrits and Van Ree, 1996*; *Pettit et al., 1984*) and the D1 antagonist SCH23390, when systemically administered, significantly decreased heroin self-administration, but had no effect when directly infused into the NAc (*Gerrits et al., 1994*).

The challenge of the DA hypothesis is also supported by genetic manipulations. For example, DA-deficient mice (targeted deletion of TH and DBH: tyrosine hydroxylase and dopamine beta-hydroxylase) still expressed conditioned place preference for morphine (*Hnasko et al., 2005*) and the downregulation of accumbal D1Rs prevented the acquisition of cocaine but not heroin self-administration (*Pisanu et al., 2015*).

If not through DA, how would opioids cause reinforcement? A model has been proposed with the pedunculopontine nucleus (PPN, called TPP in the original publication) as the initial target of opioids, which receives a descending GABA projection from the VTA. (*Bechara and van der Kooy, 1992*; *Nader et al., 1994*; *Nader and van der Kooy, 1997*). In this *scenario* DA-dependent mechanisms would take control only after chronic exposure, once dependence is established.

Not surprisingly, the question whether DA modulation underlies the reinforcing properties of opioid is therefore still hotly debated (*Badiani et al., 2011*; *Blum et al., 2015*; *Nutt et al., 2015*), which is why in the present study we use advanced genetic tools that allow for selective observation and manipulation of neuronal populations to revisit this fundamental question.

## Results

Mice were trained to intravenously self-administer heroin under a fixed-ratio one schedule (*Figure 1a*, see Methods) for 12 daily sessions of 6 hr maximum (*Figure 1b*). The dose was decreased from 50 to 25 µg/kg/infusion after six days, which led to higher acquisition rates (*Figure 1c*). The animals quickly learned to discriminate between an active and an inactive lever (after 6 days of training: 144.9 ± 26.0 active lever presses versus 8.3 ± 2.5 inactive ones; after 12 days: 283.4 ± 28 versus 20.9 ± 9.3. *Figure 1d–f*) and readily reached a robust number of heroin infusions (after 6 days of training with the higher dose: 50.6 ± 6.9 infusions; after 6 days with the lower dose: 138.1 ± 5.1 infusions after 12 days of training) in two to three hours at the end of the acquisition (*Figure 1g*). After 30d of withdrawal, mice were brought back into the apparatus in the absence of heroin injections and significantly differentiated between active and inactive lever (*Figure 1h and i*). Taken together this experiment shows that heroin was highly reinforcing and induced seeking behavior, a widely used model for relapse (*García Pardo et al., 2017*; *Shaham et al., 2003*).

To test whether heroin, when administered to drug-naïve mice, causes a DA increase in the NAc, we recorded the fluorescence changes of an intensity-based genetically encoded DA sensor (here called dLight1), in freely moving mice with fiber photometry. dLight1 enables optical readout of changes in DA concentration by coupling the agonist binding-induced conformational changes in human DA receptors to changes in the fluorescence intensity of circularly permuted (cp) GFP derived from GCaMP6 (*Patriarchi et al., 2018*). We started by replicating dopamine-specific responses in dLight-transfected HEK cells (*Figure 2a*, *Patriarchi et al., 2018*). Next, to probe DA release in freely moving mice, we delivered an adeno associated virus encoding dLight 1 (AAV9-CAG- dLight1) in the NAc, followed by implantation of an optic fiber for recordings. A group of DAT-cre mice was also injected the red-shifted opsin Chrimson (AAV8-hSyn-DIO-ChrimsonR-tdTo) into the VTA (*Figure 2b–c*). Brief (5 ms) delivery of 593 nm laser light pulses into the VTA resulted in an increase of fluorescence in the NAc that co-varied with frequency of stimulation (*Figure 2d*). To then test the effects of heroin on accumbal dopamine levels, animals were habituated to a recording arena and injected with either saline or heroin on subsequent days. Within less than a minute after the intraperitoneal heroin administration, we observed the onset of a fluorescence transient that peaked after three minutes (*Figure 2e–f*, saline: −0.008 ± 0.007, dF/F heroin: 0.133 ± 0.03 dF/F, p=0.0062, t(6) = 4.117, Paired Student's T-Test, n = 7). Importantly, the effect size was similar to that of cocaine, but was

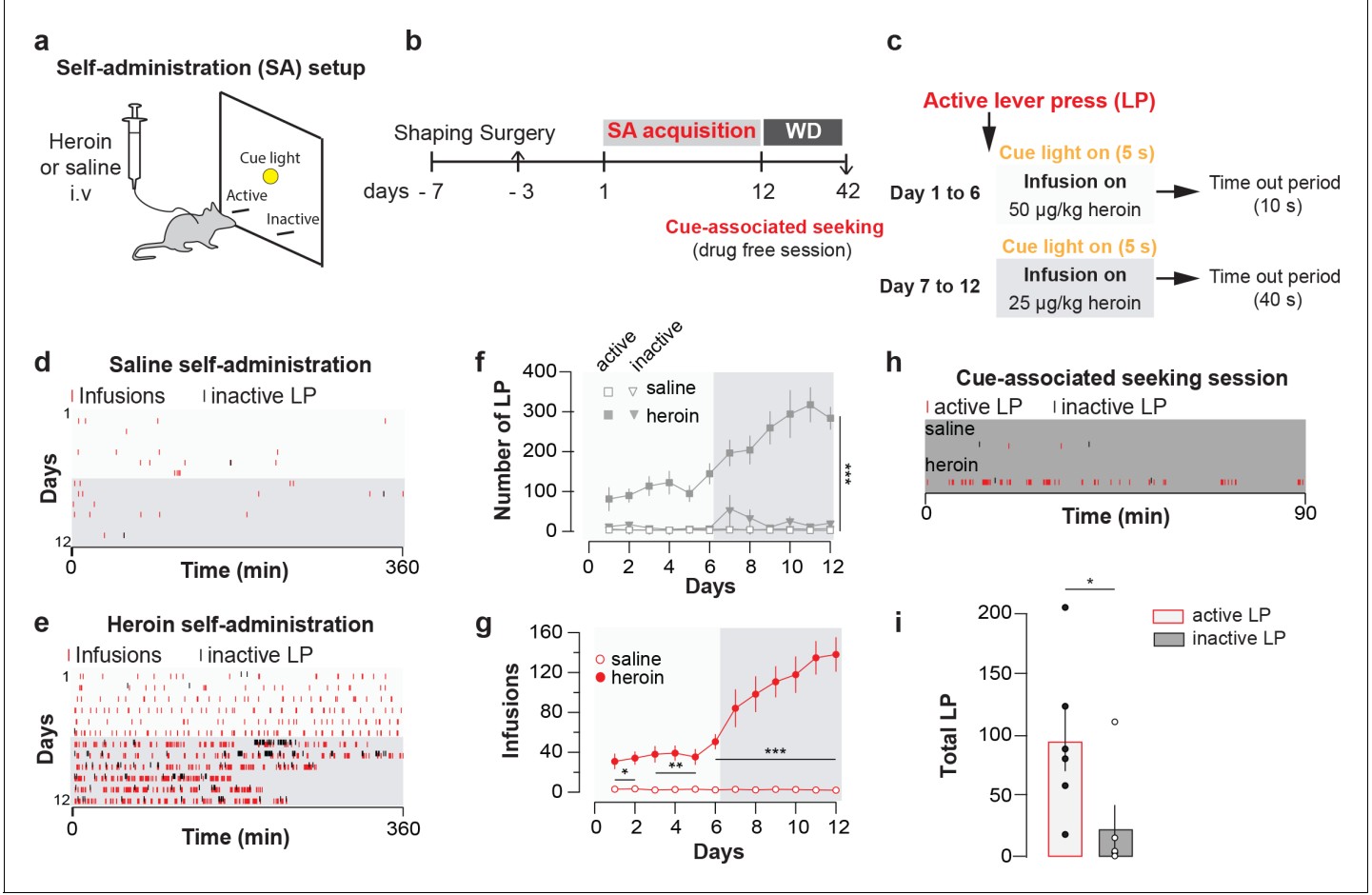

**Figure 1.** Heroin self-administration. (**a**), Schematic of behavioral setup for self-administration experiments in (**d-g**). (**b**), Day-to-day schedule of experiment for (**d–g**). (**c**), Detail of the sequence of events following a press on the active lever. An active lever press triggers the illumination of a cue-light just above the lever and an infusion of heroin. The infusion is followed by a time-out period (7 or 37 s depending on the session) where heroin is no longer available despite presses on the correct lever. WD, withdrawal. (**d**), Raster plot for infusions and inactive lever presses as a function of time during acquisition of daily self-administration session of 6 hr for a mouse that self-administered saline or (**e**), heroin. (**f**), Mean ±SEM total lever presses and (**g**), infusions during the acquisition phase of saline (n = 10) or heroin (n = 14) self-administration. Infusion rate was very robust in mice which self-administered heroin (two-way repeated-measures [RM] ANOVA, group effect, $F_{(1, 22)}$=142.2, p<0.001, time effect, $F_{(11, 242)}$=30.51, p<0.001, group X time interaction, $F_{(11, 242)}$=31.13, p<0.001; Bonferroni *post hoc* analysis, *p<0.05, **p<0.01, ***p<0.001) and animals quickly learned to discriminate between the active and inactive lever (two-way RM ANOVA, group effect, $F_{(3, 44)}$=47.16, p<0.001, time effect, $F_{(11, 484)}$=6.464, p<0.001, group X time interaction, $F_{(33, 484)}$=47.16, p<0.001; Bonferroni *post hoc* analysis, ***p<0.001). (**h**), Raster plot (top) for active and inactive lever presses as a function of time during cue-induced relapse session at day 30 of withdrawal for a mouse that self-administered either saline (top) or heroin (bottom) during acquisition phase. (**i**), Mean±SEM total lever presses at 30 days of withdrawal for mice trained for heroin (n = 6) self-administration. After 30 days of forced withdrawal seeking was robust in mice which self-administered heroin (active versus inactive lever, paired t test, $t_{10}$ = 2.31, *p<0.05).

DOI: https://doi.org/10.7554/eLife.39945.002

not seen following administration of the selective-serotonin reuptake inhibitor citalopram nor the norepinephrine reuptake inhibitor reboxetine (***Figure 2g*** saline: −0.01 ± 0.005 dF/F, citalopram: −0.008 ± 0.008 dF/F, cocaine: 0.165 ± 0.025 dF/F, reboxetine −0.009 ± 0.006 dF/F). These experiments demonstrate that dLight fluorescence specifically captures dopamine transients, and a first injection of heroin increases DA concentration in the NAc within minutes.

We then expressed a Cre-dependant GCaMP6m (AAV-DJ-EF1a-DIO-GCaMP6m) in DAT-Cre mice to record VTA DA neuron activity with fiber photometry (***Figure 2g–h***, ***Cui et al., 2013***; ***Gunaydin et al., 2014***; ***Lerner et al., 2015***) after an i.v. heroin injection in freely moving mice. Repeated infusions of heroin through the jugular vein, every 2 min, increased the calcium signal in VTA-DA neurons within minutes and a plateau is observed after the third infusion (***Figure 2i–j***, dF/F

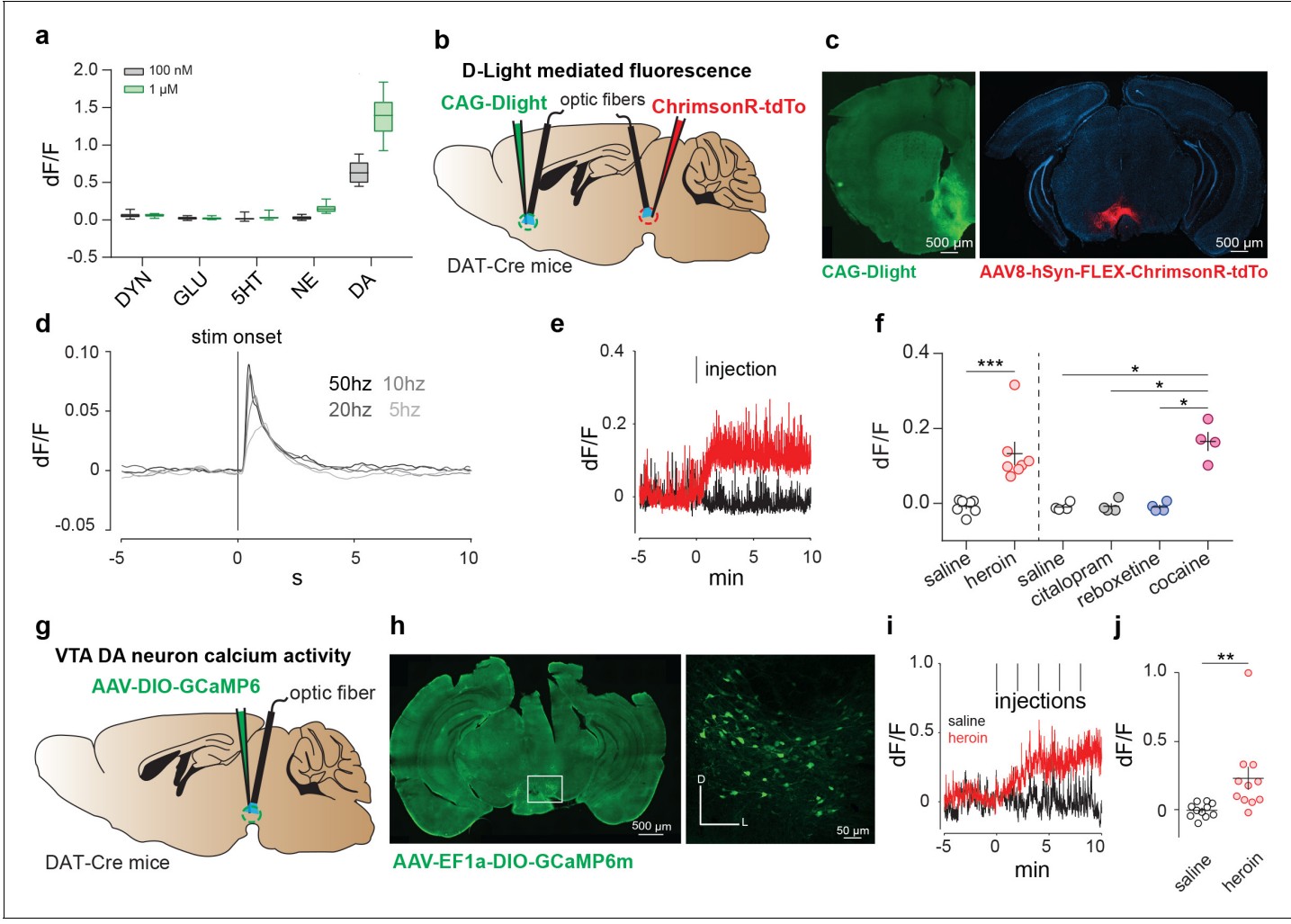

**Figure 2.** Heroin increases DA levels in the NAc via enhanced VTA DA neuron activity. (a) Fluorescence in response to five neuromodulators in HEK293 cells expressing dLight1 (DYN: dynorphin; GLU: glutamate; 5HT: serotonin; NE: norepinephrine). Data are presented as median with 25/75 percentile (box) and min-max (whiskers).( b), Schematic of the experiment for c-f; (c), Left, medial NAc shell of DAT-Cre⁺ mice were bilaterally injected with the DRD1-based DA sensor (dLight). Right, the amber light–drivable channelrhodopsin Chrimson was injected unilaterally in the VTA. (d), D-light-mediated fluorescence change following optogenetic activation of VTA DA neurons by Chrimson (mean of n = 3 animals). (e), Example trace from single animal, showing dLight-mediated fluorescence change in the NAc following intraperitoneal heroin (8 mg/kg) or saline injections. Tick mark indicates injection. (f), Average fluorescence after saline, heroin (8 mg/kg), citalopram (10 mg/kg), reboxetine (20 mg/kg) or cocaine (20 mg/kg) treatment compared to pre-infusion baseline (n = 4–7). Intraperitoneal injection of heroin or cocaine significantly increased fluorescence signals (as compared to control injections for heroin, paired t test, $t_6$ = 4.117, **p<0.01; for citalopram, reboxetine and cocaine, RM one-way ANOVA: $F_{(3,15)}$ = 42.48, p<0.01; Bonferroni post hoc analysis: *p<0.05).( g), Schematic of the experiment for h-j; h, Left, VTA of DAT-Cre⁺ mice was bilaterally injected with the floxed version of the calcium indicator GCAMP6m. Right, coronal confocal images of infected VTA. (i), Average GCaMP6m fluorescence in VTA DA neurons following first intravenous infusion of heroin (100 µg/kg/inf) or saline. Red tick marks indicate injection onset. (j), Average fluorescence after heroin or saline treatment compared to pre-infusion baseline (n = 11). Calcium transients significantly increased after heroin infusions (dF/F for saline versus heroin, Lilliefors test for normality, Wilcoxon matched-pairs signed rank test, ***p<0.001). Error bars, SEM.

DOI: https://doi.org/10.7554/eLife.39945.003

saline compared to baseline: −0.0063 ± 0.016, dF/F heroin compared to baseline: 0.22 ± 0.085, n = 11, p=0.001, Wilcoxon signed rank test). Interestingly, even though the methods are inherently technically different, the kinetic of this activity is similar to the DA surge we observed in the NAc suggesting a tight correlation between VTA DA neurons activity and DA release. Taken together, these experiments demonstrate that the first exposure to heroin in naive animals increases VTA DA neuron activity and increases DA release in the NAc.

To map the activated neurons in the VTA following heroin administration, mice were perfused after the very first self-administration session and brain slices were stained for the expression of the immediate early gene cFos and tyrosine hydroxylase (TH, *Figure 3a*). Cells positive for cFos that were also TH-immunoreactive were found most prominently in the medial part of the VTA. In this area of the VTA, 27.8% of TH$^+$ were cFos$^+$ after heroin SA, whereas 1.5% of TH$^+$ were cFos$^+$ after saline SA (*Figure 3b–e*).

DA neurons located in the medial part of the VTA preferentially project to the NAc core and medial Shell as well as in the medial prefrontal cortex, whereas DA neurons located in more lateral portions of the VTA project to the lateral Shell (*Lammel et al., 2008*). To reveal the target of the heroin-activated VTA DA neurons, we therefore injected the cholera toxin subunit B (CTB) tracer of two distinct colors (CTB-555 and CTB-488) in the medial and lateral NAc shell respectively (*Figure 3f*) confirming the topography of medio-lateral VTA-NAc parallel projections. CTB-555 seeded in the medial NAc retrogradly migrated to the medial VTA, whereas CTB-455 injected into the lateral Shell was mostly found in the lateral VTA, with very little co-localization of the two tracers (1.5%, *Figure 3g*). This result indicates that there are very few collaterals (*Yang et al., 2018*), allowing for a quantification of the co-localization of the two CTB markers with cFos after a first heroin self-administration session. We thus injected mice with the CTB tracer of two distinct colors (CTB-555 or CTB-647 in the medial or lateral NAc shell respectively, counterbalanced between animals) and submitted the mice to a session of heroin SA. We found that 51.0% of medial Shell projecting VTA neurons were also cFos positive, while this was only the case in 20.5% lateral Shell projecting cells. (*Figure 3h–j*). Taken together, heroin self-administration activates DA neurons in the medial part of the VTA that project preferentially to the medial NAc shell.

To probe for a causal relationship between enhanced mesolimbic dopamine and heroin reinforcement, we tested whether inhibiting VTA DA neurons during the initial sessions would impact acquisition and maintenance of heroin SA. To this end, we injected DAT-Cre mice expressing hSyn-DIO-hM4D(Gi) in VTA DA neurons with CNO 1 hr prior to heroin self-administration (*Figure 4a–b*). This chemogenetic intervention has been shown to be efficient to hyperpolarize DA neurons in acute midbrain slices (*Bariselli et al., 2018*). Silencing VTA DA neurons in animals where self-administration was well established significantly decreased the number of active lever presses and ensuing heroin infusions (*Figure 4c–e*, 223 ± 60 LP for 111 ± 25 infusions dropped to 23 ± 9 LP for 15 ± 6 infusions after 4 days of treatment condition (DAT-Cre$^+$ versus DAT-Cre$^-$) x CNO (present, absent), two-way repeated measures ANOVA and multiple comparison post-hoc Sidack test, *p<0.05, **p<0.005, ****p<0.0001). We further tested the necessity of VTA DA signaling during the very early heroin SA sessions. Silencing VTA DA neurons from the first to the fourth session significantly prevented the acquisition of heroin SA (*Figure 4f–h*). After CNO was stopped (from session 5), the mice quickly acquired the task and reached a number of lever presses and infusions similar to the control animals. CNO had no effect on self-administration in DAT-Cre$^-$ mice. All together these results suggest that VTA DA activity is required for the initial reinforcing properties of opioids from the very early stage of drug exposure.

We next tested whether heroin would occlude optogenetic VTA DA neuron self-stimulation as shown for cocaine (*Pascoli et al., 2015*). If heroin employs the same neuronal circuitry as VTA DA neuron self-stimulation, a heroin injection should decrease lever pressing for VTA DA activation. DAT-cre$^+$ mice were infected with the blue-light shifted excitatory opsin AAV5-DIO-hChR2-eYFP (*Figure 5a–b*). To avoid the development of tolerance to heroin, we injected various doses of heroin in a randomized order intraperitoneally (i.p.) immediately prior to the self-stimulation sessions (*Figure 5c*) and each session under heroin was followed by two sessions with free access to laser stimulation (LS) (*Figure 5c*). At baseline, the mice pressed up to 291 ± 39 times to obtain 134 ± 1.15 LS in 60 min under a fixed-ratio 1 (FR1, followed by 20 s time out period) schedule (*Figure 5d and e*). After heroin injection, the performance decreased significantly in a dose-dependent fashion (*Figure 5i*), suppressing lever pressing completely at the highest dose (*Figure 5d–i*). To rule out any sedative effects of heroin at these doses, a separate set of mice were tested over a 30 min free exploration period in an open field (see Methods). We observed that heroin actually increases locomotor activity in open field, even at the highest dose tested (*Figure 5j*). These experiments indicate that reinforcement induced by optogenetic self-stimulation of VTA DA neurons or heroin SA share underlying neural circuits.

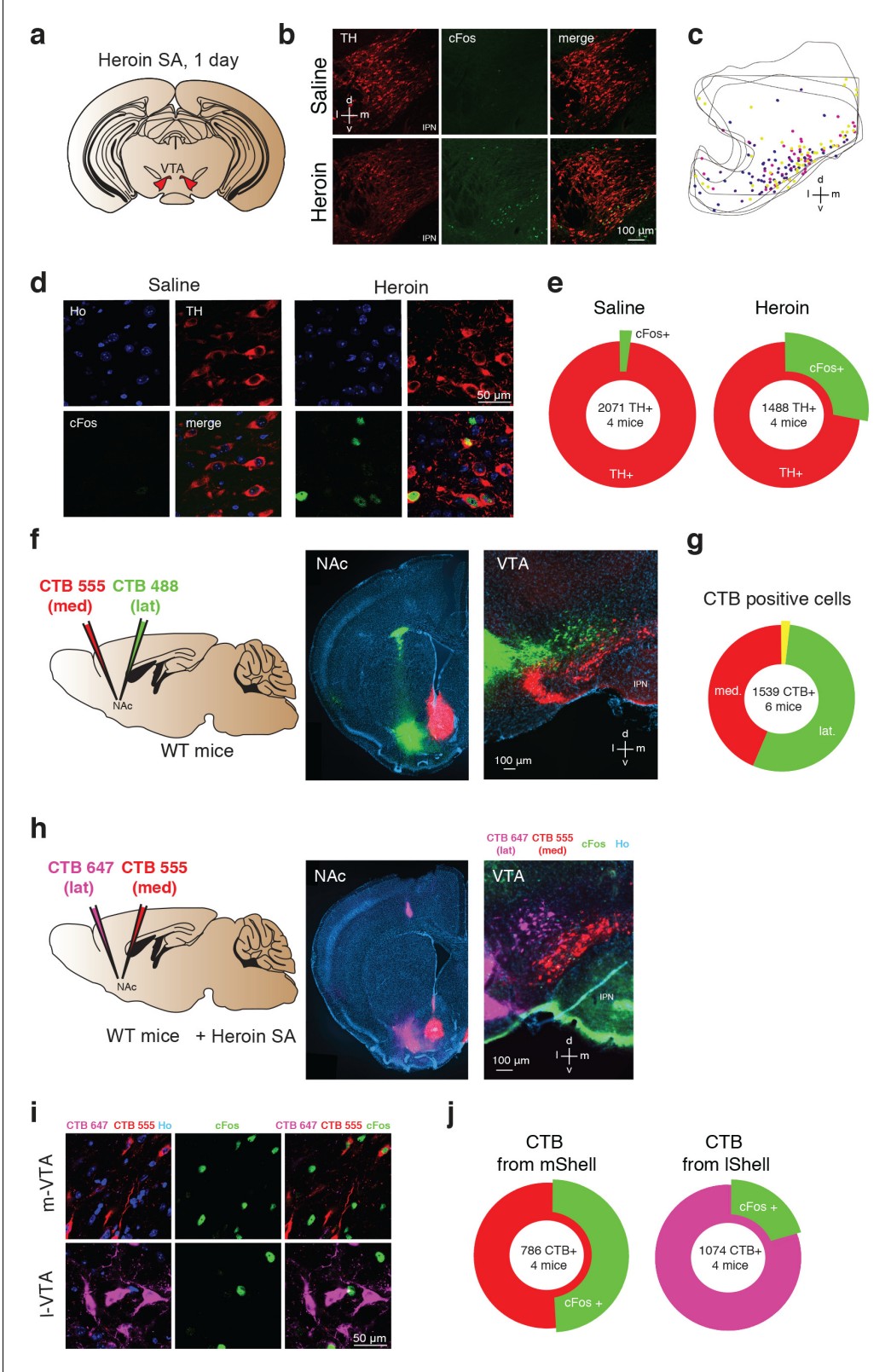

**Figure 3.** Heroin activates NAc projecting DA neurons in the medial VTA. (**a**), Schematic of experiment for (**b-e**); (**b**), TH (left, red), cFos (middle, green) staining of VTA DA neurons and co-localization of TH- and cFos-expressing neurons (right) after one day of either saline (top) or heroin (bottom) self-administration. Mice were perfused 60 min after the end of a single self-administration session. Cell nuclei are stained with Hoechst (not shown). D,

*Figure 3 continued on next page*

*Figure 3 continued*

dorsal; L, lateral; v, ventral; m, medial. (**c**) , Location within the VTA of histologically identified DA neurons expressing cFos after one day of heroin self-administration. Each color of the markers represents one animal.( **d**), High magnification confocal pictures of TH and cFos staining in saline and heroin mice. (**e**), Quantification of the TH positive VTA DA neurons also expressing cFos after one day of saline or heroin self-administration (saline: 2102 cells from four mice, heroin: 1902 cells from four mice). (**f**), Schematic of experiment for (**f–g**). The retrograde tracers CTB, conjugated to either the fluorescent dye AlexaFluor 488 (green) or AlexaFluor 555 (red) were injected in the lateral NAc shell or the medial one, respectively. In addition, a catheter implantation was performed (see methods) in order to allow heroin self-administration.( **f**), Left, coronal images showing infections in the NAc shell. Right, coronal image of the VTA. (**g**), Quantification of CTB positive cells in the VTA. (**h**), Schematic for cFos staining in the medial and lateral VTA, projecting to the NAc medial and lateral. Coronal pictures of NAc injected with CTB-555 in the medial Shell and CTB-647 in the lateral Shell and corresponding pictures in the VTA with cFos staining. (**i**), High magnification confocal images of CTB-555 and 647 with Hoechst (left), cFos (middle, green) and cFos with CTBs (right, green/red/magenta) in the medial or lateral VTA neurons after one day of heroin self-administration. Mice were perfused 60 min after the end of the self-administration session and cell nuclei have also been stained with Hoechst (not shown).( **j**), Quantification of the cFos positive VTA neurons labelled with red or magenta CTB.

DOI: https://doi.org/10.7554/eLife.39945.004

Finally to examine the involvement of VTA GABA neurons (*Tan et al., 2012*; *van Zessen et al., 2012*), we tested the reinforcing properties of their self-inhibition and asked whether heroin exposure would also occlude this behavior. We expressed the light-gated inhibitory proton pump *eArchT-3.0* in the VTA of GAD-Cre mice (*Figure 6a–b*; *O'Connor et al., 2015*) and gave the mice control over the laser switch. The mice quickly learned to discriminate the active and inactive lever by significantly increasing the number of active lever press and laser-stimulation triggered during the acquisition sessions (293.1 ± 40 LP to obtain 119 ± 15 LS in 180 min). Intraperitoneal injection of heroin just prior to the VTA GABA self-inhibition session significantly decreased the operant behavior in a dose-dependent manner (*Figure 6d–f*) and abolished this behavior at the highest dose (*Figure 6d–f*). In fact, the IC50 was very similar to the IC50 calculated based on the occlusion with DA neuron self-stimulation (6.9 vs 6.4 mg/kg, *Figure 6i* and *Figure 5i*). This experiment indicates that reinforcement by optogenetic VTA GABA self-inhibition and reinforcement by heroin share underlying neural circuits and are compatible with a disinhibitory mechanism where heroin targets GABA neurons leading to an increase of DA neurons activity.

## Discussion

In the present study, we found that heroin increases DA in the NAc through the activation of a subset of VTA DA neurons located in the medial VTA, which preferentially project to the NAc medial shell. Our chemo- and optogenetic manipulations support a disinhibitory motif and establish a link of causality with behavioral reinforcement.

The VTA consists of DA (60–65%), GABA (30–35%), glutamate (2%) and neurons that express more than one marker (*Margolis et al., 2006*; *Nair-Roberts et al., 2008*; *Roberts and Ribak, 1987*; *Steffensen et al., 1998*; *Yamaguchi et al., 2011*). Among DA neurons, subpopulations have been proposed based on DAT expression (*Blanchard et al., 1994*; *Li et al., 2013*), properties of afferent excitatory and inhibitory synaptic inputs, as well as projection to distinct targets, which have been mapped to specific functions. For example, aversive stimuli potentiate glutamatergic inputs onto DA neurons projecting to the mPFC, while rewarding stimuli potentiate inputs onto medial shell and lateral shell NAc projecting DA neurons (*Lammel et al., 2011*). The mediolateral topography of accumbens shell neurons is conserved by their dopamine inputs from the VTA, where medial and lateral shell projecting DA neurons segregate along the mediolateral VTA (*Beier et al., 2015*; *Lammel et al., 2012*; *Lammel et al., 2008*). We find that heroin during initial reinforcement preferentially activates neurons of the medial VTA projecting to the medial Shell, but cannot exclude the contribution from other projections.

Most of the afferents to midbrain DA neurons are GABAergic. Back projecting accumbal medium-spiny neurons, while targeting both DA and GABA neurons (*Beier et al., 2015*; *Henny et al., 2012*; *Yang et al., 2018*), form particularly strong synaptic connections to the latter

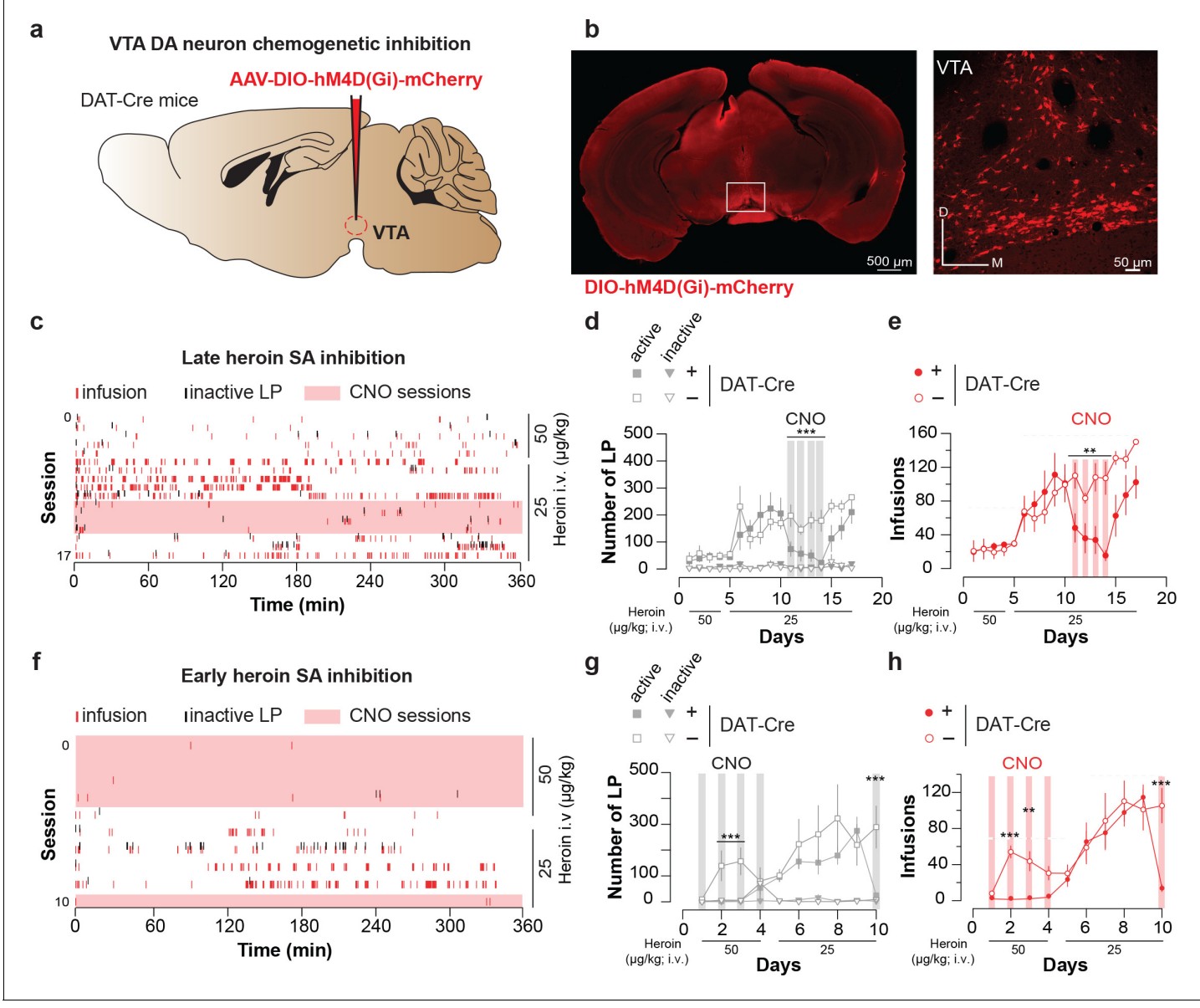

**Figure 4.** Chemogenetic inhibition of VTA DA neurons suppresses heroin self-administration. (**a**), Schematic of the experiment for (**b-e**); (**b**), Left, VTA of DTA-Cre$^+$ mice were bilaterally injected with the inhibitory DREADD hM4D. Right, coronal confocal images of infected VTA. (**c**), Raster plot for infusions and inactive lever presses during the daily acquisition sessions of heroin self-administration for a DAT-Cre$^+$ mouse. CNO (2 mg/kg) was injected intraperitoneally 20 min prior. (**d**), Mean ±SEM total lever presses and (**e**), infusions during the acquisition phase of heroin self-administration for DAT-Cre$^+$ (n = 5, closed circle) and DAT-Cre$^-$ mice (n = 6, open circle). When the self-administration behavior was well established CNO was injected prior to the session and the activation of the inhibitory DREADD dramatically decreased the number of lever presses and infusions (session highlighted in grey and red, respectively) in the DAT-Cre$^+$ animals (condition (DAT-Cre$^+$ versus DAT-Cre$^-$) x CNO (present, absent); (for LP: two-way RM ANOVA, group effect, $F_{(3, 14)}$=21.81, p<0.001, time effect, $F_{(3, 42)}$=1.366, p=0.269, group X time interaction, $F_{(9, 42)}$=1.328, p=0.252; Bonferroni *post hoc* analysis, ***p<0.001; for infusions: two-way RM ANOVA, group effect, $F_{(1, 7)}$=12.25, p<0.01, time effect, $F_{(3, 21)}$=2.664, p=0.0743, group X time interaction, $F_{(3, 21)}$=2.816, p=0.064; Bonferroni *post hoc* analysis, **p<0.01). (**f**), Raster plot for infusions and inactive lever presses during the daily acquisition session of heroin self-administration for a DAT-Cre$^+$ mouse. Twenty minutes prior to the sessions highlighted in pink, CNO (2 mg/kg) was injected intraperitoneally. (**g**), Mean ±SEM total lever presses and (**h**), infusions during the acquisition phase of heroin self-administration for DAT-Cre$^+$ (n = 6, closed circle) or DAT-Cre$^-$ mice (n = 4, open circle). CNO injection from the first session prevented the establishment of heroin self-administration behavior in the DAT-Cre$^+$ animals (condition (DAT-Cre$^+$ versus DAT-Cre$^-$) x CNO (present, absent); (for LP: two-way RM ANOVA, group effect, $F_{(3, 16)}$=37.14, p<0.0001, time effect, $F_{(4, 64)}$=3.755, p=0.0083, group X time interaction, $F_{(12, 64)}$=2.959, p=0.002; Bonferroni *post hoc* analysis, **p<0.01, ***p<0.001; for infusions: two-way RM ANOVA, group effect, $F_{(1, 8)}$=168.6, p<0.0001, time effect, $F_{(4, 32)}$=13.52, p=0.0001, group X time interaction, $F_{(4, 32)}$=8.269, p=0.0001; Bonferroni *post hoc* analysis, **p<0.01, ***p<0.001).
DOI: https://doi.org/10.7554/eLife.39945.005

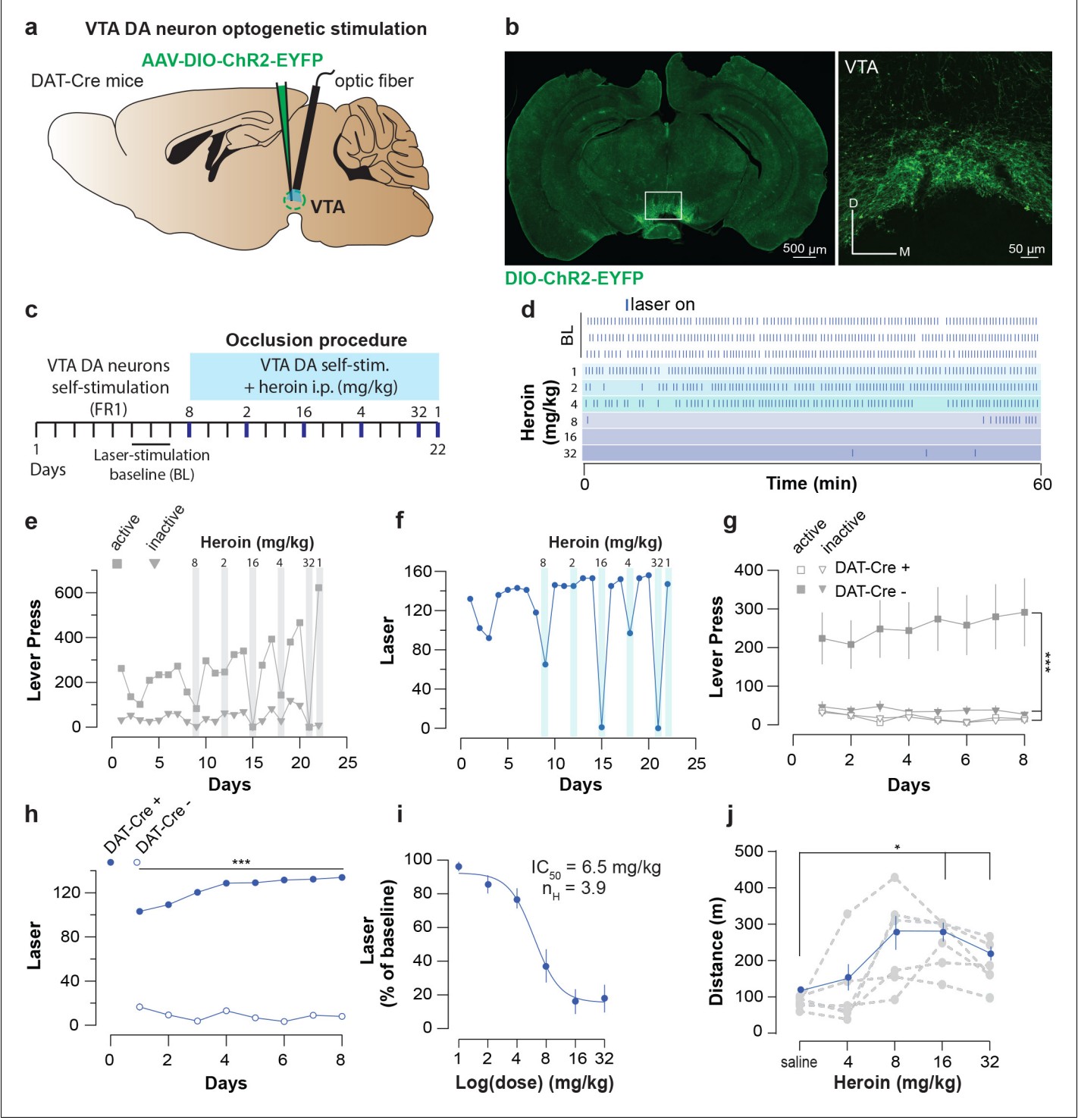

**Figure 5.** Heroin occludes optogenetic self-stimulation of VTA DA neurons. (**a**), Schematic of the experiment for (**b-f**); (**b**), Left, VTA of DAT-Cre[+] mice was bilaterally injected with a floxed version of the excitatory opsin ChR2. Right, coronal confocal images of infected VTA.(**c**), Schedule of the experiment.(**d**), Raster plot for laser stimulation during the daily acquisition session of 1 hr for a DAT-Cre[+] mouse. Right before the sessions highlighted in blue, heroin (mg/kg, dose administrated in a random order) was injected intraperitoneally. For a matter of clarity, only the three last baseline sessions are shown and the heroin sessions are arranged from the lowest dose to the highest.(**e**), Active, inactive lever presses and (**f**), laser stimulation during each session for an example DAT-Cre[+] mouse. Heroin dose-dependently reduced active lever pressing and the number of laser stimulations. (**g**), Active, inactive lever presses and (**h**), laser stimulation during the acquisition sessions of self-stimulations for either DAT-Cre[+] mice (n = 11, closed circles) or DAT-Cre[-] mice (n = 6, open circles). Establishment of self-stimulation behavior was present only in mice with expression of eYFP-ChR2 in VTA

*Figure 5 continued on next page*

Figure 5 continued

DA neurons (DAT-Cre$^+$ but not DAT-Cre$^-$ mice (for LP: two-way RM ANOVA, group effect, $F_{(3, 30)}$=38.27, p<0.001, time effect, $F_{(7, 210)}$=0.4947, p=0.8378, group X time interaction, $F_{(21, 210)}$=2.179, p=0.0029; Bonferroni *post hoc* analysis, **p<0.01, ***p<0.001; for laser stimulation: two-way RM ANOVA, group effect, $F_{(1, 15)}$=581.8, p<0.0001, time effect, $F_{(7, 105)}$=3.938, p=0.0007, group X time interaction, $F_{(7, 105)}$=8.233, p<0.0001; Bonferroni *post hoc* analysis, ***p<0.001).( i), Dose-response and fitting curve for effect of heroin i.p. injection on laser self-stimulation for DAT-Cre$^+$ (n = 11, closed circles) or DAT-Cre$^-$ (n = 6, open circles) mice. The values of IC50 and Hill coefficient are 6.5 mg/kg and 3.9 respectively. (j), Mean ±SEM of distance traveled in an open field after daily injections of increasing doses of saline or heroin (n = 6). At highest doses used (16 and 32 mg/kg), heroin significantly increased the distance traveled (saline versus heroin injection, one-way RM ANOVA, heroin doses effect, $F_{(2.475, 12.37)}$=581.80.27, p=0.00084; Bonferroni *post hoc* analysis, *p<0.05).

DOI: https://doi.org/10.7554/eLife.39945.006

(*Bocklisch et al., 2013*), which then control the activity of DA neurons (*Johnson and North, 1992*). This disinhibitory motif is particularly strong for neurons of the lateral VTA (*Yang et al., 2018*). Removal of tonic inhibition from VTA DA neurons by interneurons and accumbal projections can therefore cause increases in DA neuron activity (*Jhou et al., 2009*; *Johnson and North, 1992*) and NAc DA release (*van Zessen et al., 2012*). Using a genetically encoded DA reporter we confirm that already the very first dose of heroin increases DA in the shell.

In the alternate model of DA-independent reinforcement, the initial effect of opioids would still be on VTA GABA neurons that however mainly project to the PPN (*Laviolette et al., 2004*), a heterogeneous nucleus containing GABA, glutamate and acetylcholine neurons. The PPN projects back to the midbrain (*Oakman et al., 1995*; *Wang and Morales, 2009*; *Watabe-Uchida et al., 2012*) regulating reinforcement (*Floresco et al., 2003*; *Inglis et al., 2000*; *Lammel et al., 2012*; *Pan and Hyland, 2005*; *Steidl and Veverka, 2015*). The nearby LDT which strongly innervates the VTA may also contribute (*Omelchenko and Sesack, 2005*). There is direct evidence that the activation of the LDT-VTA pathway leads to CPP, reinforces operant responses with natural- (*Lammel et al., 2012*; *Steidl and Veverka, 2015*) and drug-rewards (*Shinohara et al., 2014*; *Wise, 2004*). However, all these scenarios converge to activate VTA DA neurons, and are thus not DA-independent. Such a 'non-dopaminergic substrate for reward within the VTA' (*Nader and van der Kooy, 1997*) is also at odds with several publications that observe an increase of extracellular DA levels in the shell following acute administration of opioids (*Aragona et al., 2008*; *Di Chiara and Imperato, 1988*; *Pontieri et al., 1995*; *Stuber et al., 2005*). Moreover our data suggest that the same circuits maintain reinforcement as exposure becomes chronic. After 12 days of heroin self-administration, inhibition of DA neurons still caused a strong, but fully reversible decrease in the responding behavior. Taken together DA-independent heroin reinforcement seems unlikely.

Heroin decreased lever-pressing for VTA DA neuron self-stimulation (*Pascoli et al., 2015*) in a dose-dependent fashion, which could not be explained by sedative effects as the same dose was able to increase locomotor activity in an open field. Such occlusion strongly suggests that heroin converges on the same cellular mechanism (similar to cocaine occlusion experiments). We also - to the best of our knowledge for the first time in the literature - observed strong reinforcement with GABA neuron self-inhibition, which was sensitive to heroin exposure. Opioids suppress the activity of VTA GABA interneurons by activation of μ-opioid receptors (MORs) (*Jalabert et al., 2011*; *Johnson and North, 1992*; *Mazei-Robison et al., 2011*), which then hyperpolarize the neurons and decrease the release probability at the axon terminal via the activation of GIRK channels and the inhibition of voltage gated calcium channels, respectively (*Cohen et al., 1992*; *Johnson and North, 1992*). The most straightforward interpretation for the effect on the behavior is thus again an occlusion scenario. Interestingly the IC$_{50}$ for this effect was virtually identical to the IC$_{50}$ observed with the occlusion of self-stimulation of DA neurons.

Our results are in direct contrast with older pharmacological experiments, where DA receptor antagonists had an effect on reinforcement of cocaine but not heroin (*Ettenberg et al., 1982*; *Pettit et al., 1984*). Maybe the receptor occupancy of the antagonists was insufficient, particularly when administered intra-cranially as suggested by the discrepancy between the systemic and intra-cranial results (*Neisewander et al., 1998*). Moreover, several studies relied on CPP rather than testing for the effect on self-administration. Moreover, validation of the pharmacological effect on neural activity in vivo remains difficult.

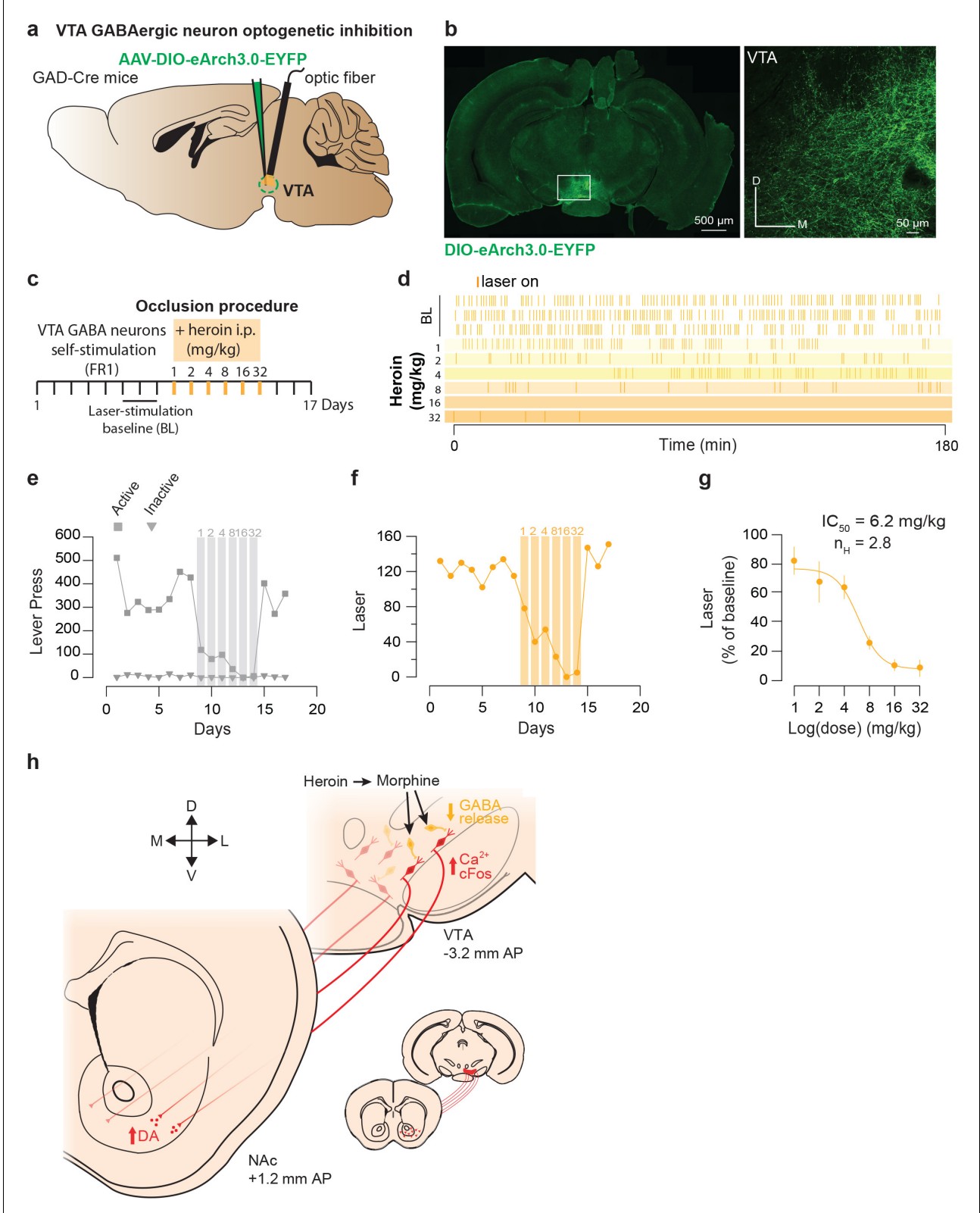

**Figure 6.** Heroin occludes reinforcing effects of self-inhibition of VTA GABA neurons. (**a**), Schematic of the experiment for **b-f**; (**b**), Left, VTA of GAD-Cre[+] mice was bilaterally injected with a floxed version of the inhibitory opsin Arch3.0. Right, coronal confocal images of infected VTA.( **c,**) Schedule of the experiment.( **d**), Raster plot for laser inhibition during the daily acquisition session of 3 hr for a GAD-Cre[+] mouse. Right before the sessions highlighted in yellow, heroin (mg/kg) was injected intraperitoneally.( **e**), total lever presses and (**f**), infusions during the acquisition phase of laser self-

*Figure 6 continued on next page*

*Figure 6 continued*

inhibition for an example GAD-Cre[+] mouse. Heroin injection resulted in a dose-dependent decrease in laser self-inhibition.( **g,**) Dose-response and fitting curve for effect of heroin i.p injection on laser self-inhibition for GAD-Cre[+] mice (n = 7). The values of IC50 and Hill coefficient are 6.2 mg/kg and 2.8 respectively. (**h,**) Summary diagram. After self-administration heroin, metabolized in morphine, binds to the MORs located on GABA neurons and activates GIRKs channels. It results in the inhibition of these neurons and the disinhibition of the DA neurons located in the ventromedial VTA. Disinhibition of these neurons leads to an increase in DA release in the medial NAc shell.

DOI: https://doi.org/10.7554/eLife.39945.007

On the other hand, there is much support for functional accumbal DA transmission underlying opioid reinforcement. Electrolytic lesions or inactivation of the NAc result in significantly decreased responding for intravenous self-administration of opioids (*Alderson et al., 2001*; *Dworkin et al., 1988*; *Suto et al., 2011*; *Zito et al., 1985*). Mice also learn to self-administer MORs agonists when infused directly into the NAc (*David and Cazala, 2000*; *Goeders et al., 1984*), most likely *via* the disinhibitory loop connecting the NAc back to the VTA. A recent study using a siRNA to downregulate D1aR in the NAc shell blocked the acquisition of cocaine but not heroin self-administration (*Pisanu et al., 2015*), raising the question for the role of specific DA receptors in drug-adaptive behavior.

The observation that morphine can induce CPP in DA-deficient mice (*Hnasko et al., 2005*) is likely explained by developmental adaptations. Moreover, these mice suffer from a severe locomotion deficit, which precluded the testing for reinforcement in a self-administration setting.

Here, by confirming the validity of the DA hypotheses for opioids, we aim to integrate work on these drugs into the emerging circuit model for addiction. Our study thus supports VTA DA neuron disinhibition for the opioid reinforcement. Untangling the circuits underlying opioid reinforcement may not only allow refining addiction treatments, but also draft the roadmap for the development of analgesic compounds without addiction liability.

## Materials and methods

**Key resources table**

| Reagent type (species) or resource | Designation | Source or reference | Identifiers | Additional information |
|---|---|---|---|---|
| Genetic reagent (*M. musculus*) | DATIRES*cre* | The Jackson Laboratory (www.jax.org) | MGI:3689434 | |
| Genetic reagent (*M. musculus*) | Gad2tm2(cre)Zjh | The Jackson Laboratory (http://www.jax.org) | MGI:4418713 | |
| Cell line (*homo sapiens, human*) | HEK293T | ATCC | Cat# CRL-1573 | |
| Recombinant DNA reagent | AAV9-CAG-dLight1.1 | Dr. Lin Tian, University of California Davis | | *Patriarchi et al., 2018* |
| Recombinant DNA reagent | AAVDJ-EF1a-DIO-GCaMP6m | Stanford Vector Core | Cat# GVVC-AAV-94 | |
| Recombinant DNA reagent | AAV8-hSyn-DIO-ChrimsonR-tdTo | UNC Vector Core | | |
| Recombinant DNA reagent | AAV5-hSyn-DIO-HM4D (Gi)-mCherry | UNC Vector Core | | |
| Recombinant DNA reagent | AAV5-EF1a-DIO-ChR2 (H134R)-eYFP | UNC Vector Core | | |

*Continued on next page*

*Continued*

| Reagent type (species) or resource | Designation | Source or reference | Identifiers | Additional information |
|---|---|---|---|---|
| Recombinant DNA reagent | AAV5-EF1a-DIO-eArch 3.0-eYFP | UNC Vector Core | | |
| Peptide, recombinant protein | CTB (Alexa FluorTM 555 Conjugate) | Invitrogen/ Thermo Fisher | Cat# C34776 | |
| Peptide, recombinant protein | CTB (Alexa FluorTM 488 Conjugate) | Invitrogen/ Thermo Fisher | Cat# C34775 | |
| Peptide, recombinant protein | CTB (Alexa FluorTM 647 Conjugate) | Invitrogen/ Thermo Fisher | Cat# C34778 | |
| Chemical compound, drug | Diacetylmorphine (heroin) | DiaMo Narcotics GmbH | DIAPHIN | |
| Chemical compound, drug | Citalopram | Cayman Chemical | Cat# 14572 | |
| Chemical compound, drug | Reboxetine | Tocris | Cat# 1982 | |
| Chemical compound, drug | Cocaine | University Hospital of Geneva | | |
| Antibody | Anti-cFos | Santa Cruz Biotechnology | RRID: AB_2106783 | (dilution 1:5000) |
| Antibody | Anti-Tyrosine Hydroxylase | Sigma-Altrich | Cat# T2928 | (dilution 1:500) |
| Antibody | Anti- GFP | Invitrogen/ Thermo Fisher | Cat# 11122 | (dilution 1:500) |
| Software, algorithm | Prism 7.02 | Graphpad | | |
| Software, algorithm | MATLAB R2017a | Mathworks | | |
| Software, algorithm | Synapse | Tucker-Davis Technologies | | |

## Animals

Wild-type C57BL/6 and transgenic mice were used throughout the study. Weights, ages and genders were homogeneously distributed among the groups. Transgenic mice were backcrossed to the C57BL/6 line for a minimum of four generations. Transgenic DAT-Cre mice (*Slc6a3*) were heterozygous BAC transgenic mice in which the Cre recombinase expression was under the control of the regulatory elements of the DA transporter gene (DAT-Cre[+] mice; (*Turiault et al., 2007*) DAT-Cre mice were originally provided by Günther Schutz. GAD-Cre[+] (Gad65Cre non-inducible;(*Kätzel et al., 2011*)) mice (*Gad2*) were also used. All animals were kept in a temperature ($21 \pm 2°C$) and humidity ($50 \pm 5\%$) controlled environment with a 12 hr light/12 hr dark cycle (lights on at 7:00). Food and water were available ad libitum, unless otherwise stated. All procedures were approved by the animal welfare committee of the Cantonal of Geneva, in accordance with Swiss law.

## Stereotaxic injections and optic fiber cannulation

Anesthesia was induced at 5% and maintained at 2.5% isoflurane (w/v) (Baxter AG) during the surgery. The animal was placed in a stereotaxic frame (Angle One) and craniotomies were performed using stereotaxic coordinates (for VTA: AP −3.3; ML −0.9 with a 10° angle; DV −4.3. For the lateral

NAc shell: AP 0.98; ML +- 1.6; DV −4.5. For the medial NAc shell: AP 1.6; ML +- 0.5; DV −4.3). Injections of virus (0.5 µl) used graduated pipettes (Drummond Scientific Company), broken back to a tip diameter of 10–15 mm, at an infusion rate of 0.05 µl / min. Following the same procedure AAV5-hSyn-DIO-HM4D(Gi)-mCherry, AAV5-EF1a-DIO-ChR2(H134R)-eYFP and AAV5-EF1a-DIO-eArch3.0-eYFP (all from University of North Carolina Vector Core) were also injected bilaterally in the VTA, while AAV-DJ-EF1a-DIO-GCaMP6m (Stanford Vector Core) and dLight1 (AAV9-CAG-dLight1.1, courtesy of Dr. Lin Tian, University of California Davis) were injected unilaterally in the VTA and the NAc, respectively. Finally, cholera toxin subunit B Alexa Fluor 555, 488 and 647 conjugate (CTB 555, CTB 488 and CTB 647, Invitrogen) were injected bilaterally in the lateral or medial NAc shell respectively. When the experimental paradigm required it, during the same surgical procedure, unique chronically indwelling optic fiber cannula (*Sparta et al., 2011*) were implanted above the VTA using the exact same coordinates as for the injection except for DV coordinate, which was reduced to 4.2. Three screws were fixed into the skull to support the implant, which was further secured with dental cement. First behavioral session typically occurs 10–14 days after surgery to allow sufficient expression of the virus.

## Implantation of jugular vein catheter

Mice were anaesthetized with a mix of ketamine (60 mg/kg, Ketalar) and xylazine (12 mg/kg, Rompun) solution. Catheters (CamCaths, model MIVSA) made of silicone elastomer tubing (outside diameter 0.63 mm, inside diameter 0.30 mm) were inserted 1.0–1.2 cm in the right jugular vein, about 5 mm from the pectoral muscle, to reach the right atrium. The other extremity of the catheter was placed subcutaneously in the mid-scapular region and connected to stainless steel tubing appearing outside the skin. Blood reflux in the tubing was checked to confirm correct placement of the catheter. Mice were allowed to recover for 3–5 days before the start of drug self-administration training and received antibiotics (Baytril 10%, 1 ml in 250 ml of water) in the drinking water during this period. Catheters were flushed daily with a heparin solution (Heparin 'Bichsel') in saline (30 IU) during the recovery period and just before and after each self-administration session.

## Self-administration apparatus

All behavioral experiments were performed during the light phase and took place in mouse operant chambers (ENV-307A-CT, Med Associates) situated in sound-attenuating cubicle (Med Associates). Two retractable levers were present on both sides of one wall of the chamber. A cue-light was located above each lever and a house light was present in each chamber. During intravenous drug self-administration sessions, the stainless steel tubing of the catheter device was connected through a CoEx PE/PVCtubing (BCOEX-T25, Instech Solomon) to a swivel (Instech Solomon) and then an infusion pump (PHM-100, Med-Associates). The apparatus was controlled and data captured using a PC running MED-PC IV (Med-Associates).

## Drug self-administration acquisition

To familiarize the mice with the operant self-administration setting, we performed four days of food shaping, whereby the mouse had to press an active lever once to obtain a food reward (FR1, one 60 min session per day, 20 mg sucrose pellet, Test Diet, USA). Following IV-catheter placement surgery mice were deprived of food for 12 hr before the first self-administration session to promote exploratory activity and were given food access ad libitum after the first session. Each session was 360 min in duration and started with the illumination of the house light and the insertion of the two levers into the operant chamber. During the first six sessions, a single press on the active lever (termed fixed-ratio one, or FR1) resulted in an infusion of 0.05 mg/kg of heroin (diacetylmorphine, DiaMo Narcotics GmbH, dissolved in 0.9% saline at 0.05 mg/mL and delivered at 0.0177 ml/s as a unit dose depending on the weight of the mouse) paired with a 5 s continuous illumination of the cue light above the active lever. Completion of the fixed-ratio also initiated a timeout period of 40 s during which heroin was no longer available. For the next six sessions, the dose of heroin was halved to 0.025 mg/kg in order to boost lever pressing (as a measure of motivation) while avoiding overdoses or any eventual sedative effect. Time out period was reduced to 10 s. The active lever (left or right lever) was randomly assigned for each mouse. To avoid an overdose of heroin, a maximum of 75 infusions for the 'high' dose and 150 for the 'low' dose were allowed per session. Only mice having

reached a stable rate of correct lever responses were included in the study. Saline control mice undertook the same procedure as heroin mice except that saline (NaCl 0.9% B. Braun) replaced heroin infusions.

## Test of cue-associated drug seeking

Thirty days after the final self-administration session (that is day 42), mice were tested for cue-associated seeking. The cue-associated drug-seeking test was a 90 min session, identical to the heroin acquisition period (house light on, insertion of the two levers), except that one press on the active lever (FR1 schedule) now triggered illumination of the cue light for 5 s but without a heroin infusion or a timeout period. The infusion pump was also activated during the drug-seeking session, because the pump noise provided an extra drug-associated cue.

## Fiber photometry cannulation and recordings

Following viral injections (see above), DAT-Cre$^+$ or wildtype mice were chronically implanted with an optic fiber (MFC_400/430–0.48_4 mm_ZF2.5(G)_FLT, Doric Lenses) above the VTA (GCaMP6m experiments) or NAc (dLight experiments). During recordings excitation (470 nm, M470F3, Thorlabs) and control LED light (405 nm, M405FP1, Thorlabs) was passed through excitation filters and focused onto a patch cord (MFP_400/430/1100–0.48_4 m_FC-ZF2.5, Doric Lenses). The fiber patch cord was connected to the chronically implanted fiber, and emission light (500–550 nm) was collected through the same fiber and passed onto a photoreceiver (Newport 2151, Doric Lenses). Excitation light was sinusoidally modulated at 211 and 531 Hz (470 nm and 405 nm light, respectively) and collected raw signal was demodulated by a real-time signal processor (RZ5P, Tucker Davis Systems) to determine contributions from 470 nm and 405 nm excitation sources (see *Lerner et al., 2015*).

For bulk GCaMP6 imaging of VTA DA calcium activity, animals were also surgically implanted with an intravenous catheter in the right jugular vein (see above). After habituation they were then recorded while freely moving in standard Med-Associates operant chambers. They were recorded for a baseline period (10 min) and then received five saline IV injections, immediately followed by a second baseline period (10 min) and five IV heroin injections. All injections were non-contingent with a two minute inter-injection interval.

For recordings of striatal dopamine dynamics using dLight, after habituation to handling, animals were injected intraperitoneally while freely moving in their homecage. In order to assess the effects of heroin (8 mg/kg), intraperitoneal Injections were performed on two experimental days separated by at least 48 hr (saline or heroin in counterbalanced design). A separate cohort of animals was subsequently treated with saline, citalopram (10 mg/kg, Cayman Chemical), reboxetine (20 mg/kg, Tocris) and cocaine (20 mg/kg) on separated recording days. During each day, fluorescence was recorded for at least a five-minute baseline period before injection, and a twenty-minute period after injection.

A subgroup of dLight animals was DAT-Cre positive (n = 3), and also injected with AAV8-hSyn-FLEX-ChrimsonR-TDTomato in the VTA, while a second optic fiber was placed above the structure. On a separate session, animals were optogenetically stimulated in the VTA while freely moving in their homecage. Using a counterbalanced sequence, bursts of 10–15 mW 593 nm wavelength light were administered once per 30 s. Bursts consisted of 5 pulses of 5 ms duration at 5, 10, 20, and 50 Hz. Laser light originated from a 593 nm DPSS laser that was gated by a shutter (CMSA-SR475_FC, Doric Lenses).

All data analyses were performed offline in Matlab (custom script https://github.com/tjd2002/tjd-shared-code, *Davidson, 2016*). To calculate dF/F, a linear fit was applied to the 405 nm signal during the baseline period to align it to the 470 nm signal, producing a fitted 405 nm signal that was used as F0 to normalize the 470 nm using standard dF/F normalization: (470 nm signal - fitted 405 nm signal)/fitted 405 nm signal. To quantify signal changes, for GCaMP6 experiments the average signal in the five minutes preceding the first IV injection (saline or heroin) was then compared to the average signal in the five minutes following the fifth IV injection (saline or heroin). For dLight experiments, the average signal in the five minutes preceding IP injection was compared to the average signal between ten and fifteen minutes after IP injection. At the end of all experiments, mice were

euthanized and brains fixed in paraformaldehyde to prepare histological slices for verification of virus expression.

## Cell culture for dLight1 sensitivity experiment

HEK293T cells (ATCC, Manassas VA #1573, STR authenticated, mycoplasma negative) were cultured and transfected as previously described (*Patriarchi et al., 2018*). Briefly, hippocampal neurons were isolated and infected using AAVs (1 x 109 GC/ml) at DIV5. Two weeks later, cells were washed with HBSS (Life Technologies) and imaged using a 40X oil-based objective on an inverted Zeiss Observer LSN710 confocal microscope with 488/513 ex/em wavelengths. For testing dLight1 sensitivity, neurotransmitters were directly applied to the bath during imaging sessions. A dual buffer gravity-driven perfusion system was used to exchange buffers between different drug concentrations. One-photon emission spectrum for the sensors was determined using the lambda-scan function of the confocal microscope. Two-photon emission spectrum was obtained with a 40X water-based objective on a SliceScore (Scientifica) and used to obtain normalized two-photon cross-section using MATLAB ROIs were generated using the threshold function in Fiji. Spatial movies and images of dF/F in response to a ligand was calculated as $\frac{\left[F(t) - \bar{F}baseline\right]}{\bar{F}baseline}$ with $F(t)$ the pixel-wise fluorescence value at each time and mean fluorescence in time points prior to ligand application, $\bar{F}baseline$.

## Immunostaining and cell counting

Mice were injected with a lethal dose of pentobarbital (150 mg/kg) and perfused trans-cardially with cold PBS and 4% paraformaldehyde solution. Brains were extracted and submerged in fixative for 24 hr at 4°C. Series of coronal 60 mm thick sections were cut on a vibratome. Immunostaining started by blocking slices in PBS 10% BSA and 0.3% Triton X-100 followed by overnight incubation in PBS 3% BSA and 0.3% Triton X-100 with primary antibody: cFos (dilution 1:5000, rabbit polyclonal, Santa Cruz, RRID: AB_2106783), TH (dilution 1:500, Mouse monoclonal anti-Tyrosine Hydroxylase, Sigma T2928) or GFP (dilution 1:500, rabbit polyclonal, Invitrogen, A11122). After three 15 min washes in PBS at room temperature, slices were incubated with 1:500 Alexa-conjugated secondary antibodies against the corresponding species (Alexa-Fluor 488, 555, Life Technologies). After three more steps of washing in PBS, a Hoechst staining was used to stain all neurons. Slices were then mounted and covered on microscope slides using mounting medium Mowiol (Calbiochem, Cat 475904–100 GM). Images were obtained in a confocal laser-scanning microscopy with a Fluoview 300 system (Olympus) using a 488 nm argon laser and a 537 nm heliumneon laser or in Leica SP5 confocal microscope using additional 350 nm laser with a 20x/0.7 NA oil immersion or objective. A semi-automated method was used to quantify viral infection and cFos expression in confocal images of brain slices containing the VTA or the NAc. Equally thresholded images were subjected to multiparticle analysis (NIH ImageJ). Region of interest (ROI) intensity values were obtained from the z stack of raw images by using Multi Measure tool. Colocalization was determined by overlap of the ROI obtained from the two independent fluorescence signals. Analysis was performed in at least three sections per animal.

## DA neuron Self-Stimulation/Inhibition Acquisition

For optogenetic studies, fiber optic cannulae of mice were connected via patch cords (Thor Labs, Germany) to a rotary joint (FRJ_1 × 2_FC-2FC; Doric Lenses, Quebec, Canada), suspended above the operant chamber. A second patch cord connected from the rotary joint to a blue or orange DPSS laser (SDL-473–100 mW or SDL-593–100 mW, respectively; Shanghai Dream Lasers; Shanghai, China) positioned outside of the cubicle. Laser power was typically 15–20 mW measured at the end of each patch cord. Thus, allowing for up to 30% power loss in connecting the patch cord to the implanted cannulae, we estimated laser power to be approximately 10–14 mW at the tip of the cannulae. In some cases, a mechanical shutter was used to control laser output (SR474 driver with SR476 shutter head; Stanford Research Systems, aligned using a connectorized mechanical shutter adaptor; Doric Lenses).

Each of the 22 optogenetic stimulation of VTA DA acquisition sessions lasted 60 min with no maximum number of reward. During all the sessions, a single press on the active lever (termed fixed ratio one, or FR1) resulted in a 10 s illumination of a cue light (pulses of 1 s at 1 Hz). After a delay of 5 s,

onset of a 15 s blue laser stimulation (473 nm) composed of 30 bursts separated by 250 ms (each burst consisted of 5 laser pulses of 4 ms pulse width at 20 Hz; *Brown et al., 2010*). A 20 s timeout followed the rewarded lever press, during which lever presses had no consequence but were recorded.

Each of the 19 optogenetic inhibition of VTA GABA acquisition sessions lasted 180 min with no maximum number of reward. During all the sessions, a single press on the active lever (termed fixed ratio one, or FR1) resulted in a 10 s illumination of a cue light (pulses of 1 s at 1 Hz). After a delay of 5 s, onset of a 15 s continuous orange laser stimulation. A 20 s timeout followed the rewarded lever press, during which lever presses had no consequence but were recorded.

## Effect of heroin on locomotor activity

To assess locomotor activity, mice were tested over a 30 min free exploration period in an open field. The apparatus consisted of Plexiglas square (40 × 40 cm). Light intensity was respectively 150 lux and 120 lux at the center and walls of the arena. Animals were injected daily with saline or increasing doses of heroin (4, 8, 16, 32 mg/kg, i.p., 10 ml/kg) immediately before being placed in the apparatus. The distance travelled was recorded and analyzed by a video-tracking system (ANY-maze; Stoelting).

## Effect of chemogenetic inhibition of VTA DA neurons on self-administration behavior

One hour before the behavioral test, DAT-Cre[+] mice expressing DREADD receptors were intraperitoneally injected with CNO 2 mg/kg in saline solution (10 ml/kg). The mice were then placed in the same settings as in a self-administration 360 min session (as described on page 12). Mice were randomly assigned to one of the two behavioral protocols. DAT-Cre[-] mice were used in the same settings as negative controls.

## Statistics

Sample sizes were calculated using publicly available sample size calculators; group sizes are in the range use for similar methodology by us and others. Experiments were typically repeated in at least two cohorts. Samples were randomly assigned to experimental groups. Experimenters were not blinded for data collection and analysis, except for cFos quantification (both acquisition and analysis),. Multiple comparisons were first subject to mixed-factor ANOVA defining both between- (for example, DAT-cre +vs DAT-cre-; saline or heroin self-administration groups) and/or within- (for example, active or inactive lever presses) group factors. Where significant main effects or interaction terms were found ($p < 0.05$), further comparisons were made by a two-tailed Student's t-test with Bonferroni correction. Single comparisons of between- or within-group measures were made by two-tailed non-paired or paired Student's t-test, respectively.

## Acknowledgements

The Swiss National Science Foundation and the European Research Council supported the work.

## Additional information

### Competing interests

Christian Lüscher: Member of the scientific advisory boards of STALICLA SA, Geneva and Phenix Foundation, Geneva. The other authors declare that no competing interests exist.

### Funding

| Funder | Grant reference number | Author |
| --- | --- | --- |
| Schweizerischer Nationalfonds zur Förderung der Wissenschaftlichen Forschung | 310030B_170266 | Christian Lüscher |
| European Commission | MeSSI | Christian Lüscher |

The funders had no role in study design, data collection and interpretation, or the decision to submit the work for publication.

## Author contributions
Julie Corre, Vincent Pascoli, Data curation, Formal analysis, Investigation, Methodology, Writing—review and editing; Ruud van Zessen, Michaël Loureiro, Data curation, Formal analysis, Writing—original draft, Writing—review and editing; Tommaso Patriarchi, Lin Tian, Resources, Writing—original draft; Christian Lüscher, Conceptualization, Supervision, Funding acquisition, Validation, Writing—original draft, Project administration, Writing—review and editing

## Author ORCIDs
Michaël Loureiro (iD) https://orcid.org/0000-0002-5915-5627
Lin Tian (iD) http://orcid.org/0000-0001-7012-6926
Christian Lüscher (iD) http://orcid.org/0000-0001-7917-4596

## Ethics
Animal experimentation: This study was performed in accordance with Swiss law (LPA). All of the animals were handled according to approved institutional animal care and use committee of Unige. The protocol was approved by the Committee on the Ethics of Animal Experiments of canton of Geneva (Permit Number: GE-128-16). Every effort was made to minimize suffering.

## Decision letter and Author response
Decision letter https://doi.org/10.7554/eLife.39945.013
Author response https://doi.org/10.7554/eLife.39945.014

# Additional files
## Supplementary files
• Supplementary file 1. Statistics table
DOI: https://doi.org/10.7554/eLife.39945.008

• Transparent reporting form
DOI: https://doi.org/10.7554/eLife.39945.009

## Data availability
The raw data are available via Zenodo (https://zenodo.org/record/1471574#.W9K7YfaYSUk).

The following dataset was generated:

| Author(s) | Year | Dataset title | Dataset URL | Database and Identifier |
|---|---|---|---|---|
| Julie Corre, Ruud van Zessen, Michaël Loureïro, Tommaso Patriarchi, Lin Tian, Vincent Pascoli, Christian Lüscher | 2018 | Dataset: Dopamine neurons projecting to medial shell of the nucleus accumbens drive heroin reinforcement. | https://dx.doi.org/10.5281/zenodo.1471574 | Zenodo, 10.5281/zenodo.1471574 |

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
