## [Decision Letter]

Thank you for submitting your article "Dopamine neurons projecting to medial shell of the nucleus accumbens drive heroin reinforcement" for consideration by *eLife*. Your article has been reviewed by three peer reviewers, one of whom is a member of our Board of Reviewing Editors and the evaluation has been overseen by a Senior Editor. The reviewers have opted to remain anonymous.

The reviewers have discussed the reviews with one another and the Reviewing Editor has drafted this decision to help you prepare a revised submission.

Summary:

The reviewers were overall positive about your study and found the manuscript balanced and scholarly in its effort to address a timely question. However, the reviewers agreed that certain key issues need to be addressed before publication. Importantly, two methods used in the study need some further validation. The authors are strongly encouraged to control for potential non-specific effects of CNO as well as establish the specificity of the dLight1 probe. The authors may already have data to address these issues. Inclusion of further controls will strengthen the arguments presented in the manuscript.

Essential revisions:

1) D-Light is a novel tool. The authors should validate the specificity of the signal. It would be great to replicate some of the controls shown in the original paper to establish the veracity of D-Light as a reporter for dopamine.

2) Given the recent issues with the selectivity and potential metabolism of clozapine-n-oxide (CNO), the authors should control the DREADD experiments with clozapine to ensure that the effects are specific to DREADD action.

---

## [Author Response]

Essential revisions:1) D-Light is a novel tool. The authors should validate the specificity of the signal. It would be great to replicate some of the controls shown in the original paper to establish the veracity of D-Light as a reporter for dopamine.

We agree with the reviewers that replication of key observations with dLight1.1 constitutes a service to the community. We have therefore carried out fluorescence measurements in HEK cells that express the dLight1.1 in response to the bath application of the major transmitters found in the NAc, confirming the molecular specificity of the sensor. Moreover, we have also looked at three additional conditions in vivo, measuring dLight1.1 fluorescence after application of different monoamine reuptake inhibitors. We find that selective serotonin reuptake inhibitor citalopram and norepinephrine reuptake inhibitor reboxetine yield no increase in the dLight1.1 signal, while cocaine administration drove a transient similar to heroin, again confirming the selectivity of the sensor.

These data have now been added to Figure 2.

2) Given the recent issues with the selectivity and potential metabolism of clozapine-n-oxide (CNO), the authors should control the DREADD experiments with clozapine to ensure that the effects are specific to DREADD action.

We thank the reviewer to bring up this important point, as CNO may be metabolized to clozapine that antagonizes endogenous DRD2 receptor weakly. For these reasons, we chose an experimental design where we compared DAT-Cre^+^ to DAT-Cre^–^ mice. In the latter the DREADD was not expressed, but they were still injected with CNO. These experiments will demonstrate whether CNO (or its metabolites) has a non-specific effect in the absence of the DREADD, which however was not the case.